# Drug Resistance in Medulloblastoma Is Driven by YB-1, ABCB1 and a Seven-Gene Drug Signature

**DOI:** 10.3390/cancers15041086

**Published:** 2023-02-08

**Authors:** Louisa Taylor, Philippa K. Wade, James E. C. Johnson, Macha Aldighieri, Sonia Morlando, Gianpiero Di Leva, Ian D. Kerr, Beth Coyle

**Affiliations:** 1Children’s Brain Tumour Research Centre, School of Medicine, University of Nottingham Biodiscovery Institute, University of Nottingham, University Park, Nottingham NG7 2RD, UK; 2Translational & Clinical Research Institute, Faculty of Medical Sciences, Newcastle University Centre for Cancer, Newcastle upon Tyne NE1 7RU, UK; 3School of Sciences, Engineering and Environment, University of Salford, Salford M5 4WT, UK; 4School of Pharmacy and Bioengineering, Keele University, Keele ST5 5BG, UK; 5School of Life Sciences, University of Nottingham, Queen’s Medical Centre, Nottingham NG7 2UH, UK

**Keywords:** medulloblastoma, drug resistance, cancer cell invasion, ABC transporter, YB-1, vincristine, metastasis

## Abstract

**Simple Summary:**

Medulloblastoma is the most common malignant childhood brain tumour. Under standard therapy, relapse occurs in 30% of patients and is almost universally fatal, accounting for 10% of all childhood cancer deaths. A barrier to effective medulloblastoma treatment is cellular resistance to standard-of-care therapies. Here, we investigate mechanisms surrounding medulloblastoma therapy resistance, both related to and independent from oncoprotein YB-1. Accordingly, we reveal roles for YB-1 in therapy sensitivity and the regulation of the multi-drug resistance gene *ABCB1*. We also identify functions for YB-1 in cell invasion, lipid metabolism and activation of the MYC oncoprotein. Importantly, through the generation of cell lines resistant to standard-of-care medulloblastoma therapies, we identify a drug-tolerant gene expression signature which may represent global, targetable mediators of medulloblastoma drug resistance. Together, our findings reveal important mechanisms and genes underlying therapy resistance in medulloblastoma and provide routes to their intervention.

**Abstract:**

Therapy resistance represents an unmet challenge in the treatment of medulloblastoma. Accordingly, the identification of targets that mark drug-resistant cell populations, or drive the proliferation of resistant cells, may improve treatment strategies. To address this, we undertook a targeted approach focused on the multi-functional transcription factor YB-1. Genetic knockdown of YB-1 in Group 3 medulloblastoma cell lines diminished cell invasion in 3D in vitro assays and increased sensitivity to standard-of-care chemotherapeutic vincristine and anti-cancer agents panobinostat and JQ1. For vincristine, this occurred in part by YB-1-mediated transcriptional regulation of multi-drug resistance gene *ABCB1*, as determined by chromatin immunoprecipitation. Whole transcriptome sequencing of YB-1 knockdown cells identified a role for YB-1 in the regulation of tumourigenic processes, including lipid metabolism, cell death and survival and MYC and mTOR pathways. Stable cisplatin- and vincristine-tolerant Group 3 and SHH cell lines were generated to identify additional mechanisms driving resistance to standard-of-care medulloblastoma therapy. Next-generation sequencing revealed a vastly different transcriptomic landscape following chronic drug exposure, including a drug-tolerant seven-gene expression signature, common to all sequenced drug-tolerant cell lines, representing therapeutically targetable genes implicated in the acquisition of drug tolerance. Our findings provide significant insight into mechanisms and genes underlying therapy resistance in medulloblastoma.

## 1. Introduction

Medulloblastoma represents the most frequent malignant paediatric brain tumour. It comprises four principal molecular groups with different patterns of metastasis and overall prognosis—WNT (MB_WNT_), SHH (Sonic Hedgehog; MB_SHH_), Group 3 (MB_Group3_) and Group 4 (MB_Group4_)—which can now be further categorised into second-generation sub-groups [1,2]. Under conventional treatment regimens (neurosurgery, craniospinal irradiation and chemotherapy), approximately 30% of patients will relapse, an occurrence which is almost universally fatal [3]. The prognosis for MB_Group3_ is particularly poor due to the frequent occurrence of metastasis in this group, MB_SHH_ and MB_Group4_ have intermediate outcomes, whereas the outcome for MB_WNT_ is generally good. Accordingly, improving our understanding of resistance mechanisms surrounding standard-of-care medulloblastoma chemotherapy represents an important area of medulloblastoma research [4], and will drive future improvements in therapy.

Y-box binding protein 1 (YB-1), is a multi-functional protein encoded by the *YBX1* gene on chromosome 1p34.2. Although originally identified as a transcription factor, YB-1 has since been implicated in almost all mRNA- and DNA-dependent processes in the cell, with recorded roles in mRNA translation and packaging, DNA repair, proliferation, pre-mRNA splicing and DNA replication (reviewed extensively in [5]). Over-expression has been described in numerous malignancies including renal cell carcinoma [6], breast cancer [7], osteosarcoma [8], head and neck squamous cell carcinoma [9], prostate cancer [10], glioma [11] and non-small cell lung cancer [12]. Frequently, elevated expression strongly correlates with cancer progression, poor prognosis, aggressive disease, increased metastatic potential and importantly, chemoresistance (reviewed in [13]). In tumour-derived cell lines, YB-1 knockdown has been shown to increase cellular sensitivity to numerous cytotoxic drugs including cisplatin, etoposide, temozolomide and taxol [12,13,14]. Accordingly, YB-1 is linked to multiple cellular resistance mechanisms. YB-1 can preferentially bind cisplatin-modified, abasic and mismatched DNA and exhibits exo- and endo-nuclease activity at such sites [15,16], whilst also interacting with multiple members of base- and nucleotide-excision repair systems [17,18,19]. YB-1 has also been reported to disrupt p53-mediated apoptosis, both by transcriptional repression of TP53, BAX (Bcl2-associated X protein) and NOXA (NADPH oxidase activator) [20,21] and through YB-1-mediated activation of MDM2 and subsequent P53 degradation [22], further highlighting a role for YB-1 in DNA repair and the evasion of genotoxic stress-induced apoptosis. Notably, several laboratories have reported an association between nuclear YB-1 expression and the expression of the multidrug pump *ATP binding cassette subfamily B member 1* (*ABCB1*) following therapeutic exposure, linking YB-1 expression to multidrug resistance mechanisms [8,23].

The aforementioned studies highlight clear YB-1 involvement in cellular mechanisms surrounding stress response and drug resistance in multiple tumour types. However, how YB-1 is associated with such mechanisms in medulloblastoma remains to be explored. Through the creation of YB-1 knockdown MB_Group3_ cell lines and subsequent transcriptomic analyses, the current study aimed to investigate the role of YB-1 in medulloblastoma tumourigenesis, with a focus on drug resistance. Bioinformatic analysis of publicly available medulloblastoma patient datasets was utilised to validate pre-clinical findings, whilst in vitro 3D invasion, cytotoxicity and chromatin immunoprecipitation assays provided key functional validation. Alongside this, we designed a global approach to identify alternative mechanisms driving resistance to standard-of-care therapies. Vincristine- and cisplatin-tolerant MB_Group3_ and MB_SHH_ cell lines were derived and mechanisms driving therapy resistance were identified by next-generation sequencing. Our findings, and the tools that we have created, identify key mediators of medulloblastoma drug resistance and routes to their intervention.

## 2. Materials and Methods

### 2.1. Cell Lines and Standard Culture Conditions

DAOY and D283 cell lines were obtained from ATCC (Manassas, USA), ONS76 from Annette Künkele (Charité Universitätsmedizin, Berlin, Germany), UW-228-3 and D458 from John R. Silber (University of Washington, Seattle, USA), HD-MB03 from Till Milde (DKFZ, Heidelberg, Germany) and CHLA-01 and CHLA-01R from Geoff Pilkington (University of Portsmouth, UK). DAOY, D283, D425, and D458 cells were cultured in DMEM with 10% foetal bovine serum (FBS, HyClone (Logan, Utah, USA), SH30541.03). UW-228-3 cells in DMEM/F-12 with 15% FBS and 1% sodium pyruvate. ONS76 and HD-MB03 cells were cultured in RPMI 1640 with 10% FBS. CHLA-01 and CHLA-01R were cultured in DMEM/F-12 supplemented with 2% B-27 (Gibco (Loughborough, UK), 17504044), 20 ng/mL recombinant human epidermal growth factor (EGF, Gibco, PHG0315) and 20 ng/mL recombinant human basic fibroblast growth factor (bFGF, Gibco, PHG0266). All cell lines were grown under antibiotic-free culture conditions at 5% CO_2_ and 37 °C. Mycoplasma testing was performed monthly using a PlasmoTest^TM^ Mycoplasma Detection kit (InvivoGen (San Diego, CA, USA); rep-pt1) as per the manufacturer’s instructions.

### 2.2. Cytotoxicity Assays

For drug response analysis, cells were grown in standard culture conditions as previously described and harvested when they reached approximately 70% confluence. Cells were seeded at a density of 10,000 cells/well (D283), 5000 cells/well (HD-MB03 and D425) or 1000 cells/well (DAOY) in black-walled, clear-bottom 96-well plates and left to settle for 24 h. Cells were treated with cisplatin (Selleckchem (Houston, TX, USA), S1166), vincristine (Selleckchem, S1241), JQ1 (Selleckchem, S7110), panobinostat (Selleckchem, S1030) or an equivalent concentration of drug vehicle (dimethyl formamide (DMF) or dimethyl sulfoxide (DMSO))—and incubated at 37 °C and 5% CO_2_ for 72 h. Following treatment, surviving cells were assayed with PrestoBlue (ThermoFisher (Waltham, MA, USA), A13262) at a final dilution of 1:10 for 60 min at 37 °C and 5% CO_2_ and fluorescence was measured at 560/590 nm using a FLUOstar Omega (BMG Labtech (Ortenberg, Germany)) microplate reader. The drug response was calculated as a percentage of the vehicle-treated control and dose–response curves and IC_50_ values were generated by non-linear regression analysis using GraphPad Prism 8 (GraphPad Software Inc., La Jolla, CA, USA).

### 2.3. Establishment of Drug-Tolerant Cell Lines

To generate drug-tolerant cell lines a continuous model of selection was utilised by which D283, D458 and HD-MB03 cell lines were cultured continually in the presence of cisplatin, and the DAOY cell line in the presence of vincristine, in antibiotic-free media at 5% CO_2_ and 37 °C and the treatment dose escalated upon cell proliferation. Cells were passaged in T-25 flasks in duplicate and dosing was commenced at 1/100 IC_50_ in each cell line. Matched DMF-treated (DMSO-treated for DAOY) vehicle control flasks were passaged bi-weekly alongside drug-treated flasks to account for morphological and genetic changes occurring from long-term culture and vehicle exposure. Drug resistance was measured by drug response assays, with cells considered resistant when the IC_50_ value of the treated cells had exceeded the treatment dose and the cell lines exhibited a significant increase in fold resistance (fold resistance = IC_50_ of tolerant cell line/IC_50_ of parental cell line).

### 2.4. Quantitative Real-Time PCR Analysis

RNA isolation of cell lines was performed using the NucleoSpin^®^ RNA Plus kit (Machery-Nagel (Allentown, PA, USA); 740984). RNA was converted to cDNA using superscript II reverse transcriptase (Invitrogen (Waltham, MA, USA); 1080-044). Gene expression of the resultant cDNA template was assessed by quantitative reverse transcription PCR (CFX384 RT-PCR machine; BIORAD (Hercules, CA, USA)) using iQ SYBR SuperMix (BIORAD; 1708884). Primer sequences were utilised as follows: *YB-1* forward (5′ AAG AAG GTC ATC GCA ACG AAG 3′) and *YB-1* reverse (5′ CTC CTA CAC TGC GAA GGT ACT 3′); *ABCB1* forward (5′ CCC ATC ATT GCA ATA GCA GG 3′) and *ABCB1* reverse (5′ GTT CAA ACT TCT GCT CCT GA 3′); *ABCC1* forward (5′ TTC TCG GAA ACC ATC CAC GA 3′) and *ABCC1* reverse (5′ CCT GTG ATC CAC CAG AAG GT 3′); *GAPDH* forward (5′ ATG TTC GTC ATG GGT GTG AA 3′) and *GAPDH* reverse (5′ CTC TTC TGG GTG GCA GTG AT 3′). The housekeeping gene *GAPDH* was used as a control to normalize the data and the relative mRNA expression level was calculated using ΔC_q_/ΔΔC_q_ methodology [14].

### 2.5. Western Blotting

Cells were lysed in NP-40 lysis buffer containing 1 × cOmplete™ EDTA-free Protease Inhibitor Cocktail (11836170001, Roche (Basel, Switzerland)) on ice for 30 min with regular vortexing. Lysates were centrifuged at 14,000× *g* for 15 min at 4 °C and the resultant supernatant was harvested. The supernatant was mixed with 4 × SDS loading buffer and boiled at 95 °C for 10 min and subjected to SDS-PAGE on 12% SDS-PAGE gels. Following electro-blotting, samples were probed with the following antibodies: anti-YB-1 at 1:1000 (Cell Signalling Technology (Danvers, MA, USA); 4202S); anti-GAPDH at 1:1000 (Cell Signalling Technology; 2118S); HRP-linked anti-Rabbit IgG at 1:2000 (Cell Signalling Technology; 7074S). To visualise proteins, membranes were incubated with Pierce™ ECL Western Blotting Substrate (Thermo Fisher Scientific (Waltham, MA, USA); 32106). Chemiluminescence was then measured using a LAS Mini 3000.

### 2.6. Modified Transwell Invasion Assay

Prior to assay commencement, cells were starved in reduced-serum (2% FBS) media, and 8 µm 24-well plate transwell inserts (Greiner (Kremsmünster, Austria); 662641) were coated with 50 µL 10 µg/mL Cultrex^®^ mouse collagen IV (Bio-Techne (Abingdon, UK) 3410-010-02). Following 24 h, 50 µL of 100 µg/mL Laminin I (Cultrex; 3400-010-02) was added to each coated insert, which was partially dried for 1 h. 1 × 10^5^ viable cells were then plated in serum-free media within the coated inserts and the outer chamber was filled with complete medium (10% FBS). Plates were incubated for 48 h at 37 °C and 5% CO_2_. Media was then removed from the chambers and invaded cells dislodged from the bottom surface of the insert by the addition of 1× cell dissociation solution for 1 h at 37 °C and 5% CO_2_ (AMS-Bio (Abingdon, UK); 3455-05-03). A PrestoBlue cell viability assay was used as previously described to assess metabolic activity and quantify cell migration relative to a standard curve of known cell numbers.

### 2.7. shRNA Transduction

Cell lines with stable knockdown of YB-1 expression were generated through shRNA-mediated gene silencing using the GIPZ Lentiviral particle starter kit for *YBX1* (Horizon Discovery (Waterbeach, UK); VGH5526-EG4904) according to the manufacturer’s instructions.

### 2.8. Chromatin Immunoprecipitation

D283 and HD-MB03 cells were cross-linked in 1% paraformaldehyde for 10 min. Cell lysis, shearing, immunoprecipitation and DNA purification were undertaken using a Magna ChIP A/G immunoprecipitation kit (Sigma Aldrich (St. Louis, MO, USA); 17-10085) according to the manufacturer’s instructions. Samples were sonicated using a water bath sonicator (Diagenode, Denville, NJ, USA) to obtain 200–500 bp chromatin fragments. The following antibodies were utilised to pull down protein-DNA complexes, with Immunoglobin (IgG) 1 used as a negative control and histone H3K4Me3 used to confirm active promoter regions: YB-1 (Santa Cruz (Santa Cruz, CA, USA); SC101198); IgG1 (Cell Signalling; 5415); H3K4me3 (Active Motif (Carlsbad, CA, USA); 61379). To quantify target protein interaction with the gene of interest, qPCR using primers specific to an inverted CCAAT box within the *ABCB1* promoter region was undertaken. Primer sequences were as follows: *ABCB1* forward (5′ CAT GCT GAA GAA AGA CCA CTG C 3′) and *ABCB1* reverse (5′ AGG CTT CCT GTG GCA AAG AG 3′).

### 2.9. Whole Transcriptome Sequencing

YB-1 knockdown and non-silencing control lines were pelleted in triplicate in consecutive passages and RNA was extracted using the RNeasy Mini kit (Qiagen; 74104) according to the manufacturer’s instructions. Whole transcriptome sequencing was performed by Qiagen Genomic Services (Hilden, Germany). Library preparation was undertaken using the QIAseq Stranded Total RNA Library Kit (Qiagen; 180745) with QIAseq FastSelect rRNA/globin depletion according to the manufacturer’s instructions. Libraries were sequenced on a NextSeq 500 instrument (Illumina (San Diego, CA, USA); SY-415-1002) in a dual index 1 × 75 bp format at a sequencing depth of 30 M reads/sample. Sequencing data analysis was conducted using CLC Genomics Workbench (Qiagen; version 12.0.2) and CLC Genomics Server (Qiagen; version 11.0.2). Differentially expressed genes (DEGs) were defined as log_2_ fold change ≥0.5 or ≤−0.5, *p*-value ≤ 0.05 and False Discovery Rate (FDR) *p*-value ≤ 0.1 (Appendix A). Ingenuity Pathway Analysis (IPA) software (Qiagen) was used for subsequent bioinformatics analysis, which included canonical pathway, disease and function and upstream regulator analysis. In each case, significance was set at *p* < 0.05 with the Benjamini-Hochberg (B-H) method employed to control FDR. Where possible, Z-score activation predictions were calculated. Z-scores > 2 or <−2 were considered as significantly activated or inhibited, respectively.

### 2.10. 3′mRNA Sequencing

Drug-tolerant cell lines were pelleted in triplicate in consecutive passages and RNA was extracted as previously described. Libraries were prepared using the QuantSeq 3′mRNA-Seq library prep kit for Illumina in the forward read direction supplemented with unique molecular index (UMI) as per manufacturer’s instructions (Lexogen (Vienna, Austria); 015.96). The resultant library pool was sequenced on a NextSeq 500 High Output v2.5 75 cycle kit (Illumina (San Diego, CA, USA); 20024906), to generate approximately 5 million 75 bp single-end reads per sample. Differential gene expression analysis was undertaken using the R statistical environment package DESeq2 (version 1.24.0) with default settings. DEGs were defined as Log_2_ fold change ≥0.5 or ≤−0.5 and B-H-adjusted *p*-value ≤ 0.05. Gene ontology enrichment analysis of differentially expressed genes was conducted with R package GOSeq (version 1.4.0) with default settings and a cut-off *p*-value of *p* ≤ 0.05.

### 2.11. Statistical Analysis

Results are shown as mean ± SEM of the indicated number of independent experiments. The statistical significance of differences in group results was compared using one- or two-way analysis of variance (ANOVA) with multiple comparison testing as indicated. All statistical analyses and plots were carried out using GraphPad Prism 8 (GraphPad Software Inc., La Jolla, CA, USA) unless stated otherwise. The number of biological samples, corresponding statistical test and significance levels are indicated in each figure legend.

### 2.12. Bioinformatic Analysis of Published Datasets

Published medulloblastoma patient datasets as described by Cavalli et al. [15], Northcott et al. [16], Wang et al. [17] and Weishaupt et al. [18] were accessed and analysed using the R2: Genomics Analysis and Visualization Platform (http://r2.amc.nl (accessed on 6 April 2020)). A cut-off between high YB-1 gene expression and low YB-1 gene expression groups was selected, where *p*-values obtained from the log-rank test were minimized.

## 3. Results

### 3.1. High YB-1 Expression Correlates with Poor Overall Survival in Medulloblastoma

As little is known regarding the functional role of YB-1 in medulloblastoma, YB-1 gene expression was first studied across a published human medulloblastoma cohort comprising >200 samples [16]. YB-1 gene expression was found to be elevated across all four principal medulloblastoma groups relative to normal human cerebellum control samples (Figure 1A). We then investigated a correlation between candidate gene expression and prognosis in another published cohort of 766 patients [15]. MB_Group3_, MB_Group4_ and MB_SHH_ patients with high expression levels of YB-1 mRNA had significantly worse 10-year overall survival outcomes than those with low YB-1 expression levels (Figure 1B–D), indicative of YB-1 oncogenic potential in medulloblastoma. The same trend was not observed in MB_WNT_ tumours, where no correlation between YB-1 expression and survival was detected (Appendix A). Next, in the same large-scale patient cohort, the relationship between high YB-1 expression and metastasis was explored [15]. Patients who presented with metastatic disease exhibited significantly higher YB-1 than non-metastatic patients (Appendix A). Finally, in a smaller published cohort comprising matched primary and metastatic tumour samples [17], we observed that 7/9 patients had elevated YB-1 expression in their metastatic tumour compared to their matched primary tumour, further indicating potential involvement of YB-1 in the metastatic cascade (Appendix A).

Analysis of protein and gene expression in established and validated cell line models of MB_SHH_, MB_Group3_ and MB_Group4_ demonstrated universal YB-1 mRNA and protein expression (Figure 1E,F; Appendix A) [19,20,21,22,23,24,25]. Subsequent experiments were conducted in MB_Group3_ models on account of the propensity of MB_Group3_ patients to relapse [3], combined with the high expression level of YB-1 mRNA and protein detected across MB_Group3_ cell lines and patient samples.

### 3.2. YB-1 Depletion Impedes the Invasive Capability of Medulloblastoma Cells

Genomic analysis of medulloblastoma patient datasets highlighted the potential association between YB-1 expression and metastatic medulloblastoma (Appendix A). To explore this further, YB-1 mRNA expression was depleted in D283 and HD-MB03 cell lines by way of shRNA lentiviral transduction. Each cell line was transduced with either a non-silencing control (NS) or one of three YB-1 shRNA constructs (Y_A, Y_B and Y_C) for 72 h. Only the Y_A shRNA construct resulted in sustained and significant YB-1 mRNA depletion in both D283 and HD-MB03 cells (Appendix A) and thus was taken forward for further study. Expression was reduced by 93% and 83% in the KD-HD-MB03-Y_A line (KD-HD-MB03) and 73% and 50% in the KD-D283-Y_A line (KD-D283) at an mRNA and protein level, respectively (Figure 2A,B; Appendix A).

To assess if such gene expression changes following YB-1 knockdown resulted in functional alterations in medulloblastoma cell migration and invasion, a modified Boyden chamber assay was utilised, designed to recapitulate invasion through a simplified brain extracellular matrix (ECM)-like barrier. Cells were seeded in serum-free media in the upper chamber of the assay and incubated for 48 h to facilitate migration (uncoated membrane), or invasion (coated membrane), towards a complete media-containing lower chamber (Figure 2C). Abundant components of the brain ECM—collagen IV and laminin 1—were selected to form the membrane coating. YB-1 knockdown was found to significantly decrease the number of invaded cells in both KD-HD-MB03 and KD-D283 lines compared to non-silencing control lines (NS-D283 and NS-HD-MB03), with an observed 36% and 46% reduction in cell invasion, respectively (Figure 2E). Given that YB-1 knockdown does not significantly impede proliferation in either cell line over a 72 h time course (Figure 2F), we conclude that the observed reduction in invaded cells arises from decreased invasive capability following YB-1 depletion and not reduced cell growth.

Comparatively, no alteration in migration was observed upon YB-1 knockdown in either cell line (Figure 2D), indicating that YB-1 predominantly regulates cellular traits associated with invasion through the laminin 1/collagen IV coating, rather than simple migration through the transwell insert. As YB-1 has been associated with the regulation of matrix metalloproteinases (MMP) MMP-2 and MMP-9 [12,26], zymography was undertaken to assess if the observed decrease in cellular invasion arose from decreased MMP activity. No alteration in MMP-2 and MMP-9 expression or activity was observed, suggesting the involvement of alternative invasion mechanisms (Appendix A).

### 3.3. YB-1 Functions in Numerous Key Cellular Pathways in Medulloblastoma Cells

We next wanted to build a picture of the global YB-1 transcriptome in Group 3 medulloblastoma. Accordingly, we performed whole transcriptome sequencing on each YB-1 knockdown line and the appropriate non-silencing control line. Analysis of the most significantly altered genes (up and down) confirmed YB-1 to be down-regulated in both cell lines (Figure 3A,B; top 10 differentially expressed genes (DEGs) are named), providing an internal control for the shRNA-mediated knockdown. No other genes were identified as common between D283 and HD-MB03 cell lines, indicative of differential genetic responses following YB-1 genetic manipulation, despite the MB_Group3_ status of each line. Notable top 10 DEGs included those encoding extracellular matrix components *COL8A1* and *TNC* and cell adhesion molecule *CADM3*, which have previously been associated with cellular invasion and migration in gastric cancer [27] and glioma [28,29] respectively, further supporting a function for YB-1 in the regulation of invasion/migration processes in medulloblastoma cells. Unsupervised hierarchical clustering of the top 50 most altered genes across YB-1 knockdown and non-silencing samples can be found in Appendix A.

Given that any gene is likely to be part of a more complex biological process, Ingenuity Pathway Analysis (IPA) was employed to predict affected cellular biology from statistically significant differential gene expression patterns. Despite no commonality at a gene level, analysis of affected molecular and cellular functions identified cell death and survival processes to be dysregulated in both cell lines following YB-1 knockdown (Figure 3C,D). Examination of the most significantly altered canonical pathways in each cell line supported this, revealing significant inhibition of the sirtuin signalling pathway (Z score = −2) in the KD-HD-MB03 line, a stress response pathway with emerging roles in tumourigenesis, multidrug resistance (reviewed in [30,31]) and medulloblastoma [32,33] (Figure 3E). These observations were validated by IPA of a published medulloblastoma patient cohort [15] separated by high-low YB-1 gene expression, in which significant dysregulation in a number of cell death and survival processes was also detected (Appendix A).

IPA molecular and cellular functions analysis also revealed enrichment for metabolic processes, with alterations in carbohydrate metabolism specific to HD-MB03, vitamin and mineral metabolism specific to D283 and lipid metabolism affected across both cell lines, (Figure 3C,D). Examination of the most significantly altered canonical pathways in each cell line corroborated these findings, revealing a predominance of altered lipid and cholesterol biosynthesis pathways following YB-1 knockdown (Figure 3E). Palmitate Biosynthesis I, Fatty Acid Biosynthesis Initiation II and Stearate Biosynthesis I was significantly altered in the KD-HD-MB03 line, while the Superpathway of Cholesterol Biosynthesis, Mevalonate Pathway I and Superpathway of Geranylgeranyl Diphosphate (GGPP) were significantly altered in the KD-D283 line. Canonical pathway analysis of the aforementioned publicly available human medulloblastoma cohort separated by high-low YB-1 expression also identified significant inactivation of the Cholesterol Biosynthesis Pathway (Appendix A), highlighting a potentially novel function for YB-1 in the regulation of lipid metabolism in brain tumour cells.

To further analyse the YB-1 transcriptome in Group 3 medulloblastoma, we also performed upstream regulator analysis, a subset of IPA that allows the identification of upstream regulators that may be responsible for the gene expression changes in a particular dataset (Figure 3F). The benefit of upstream regulator analysis is that rather than simply assessing regulator gene expression, the analysis pipeline uses gene expression changes of downstream targets to estimate the regulator’s functional state. In this way, we identified the MYC oncoprotein to be significantly inactivated, both in D283 cells following YB-1 knockdown (Z score = −2.1) and in medulloblastoma patients with low YB-1 gene expression (Appendix A). MYC protein expression was unaltered in both YB-1 knockdown cell lines (Louisa Taylor, Ian Kerr & Beth Coyle, University of Nottingham, UK, observation, 2021) indicating that this inhibitory effect occurs at an activity level, a finding with potential therapeutic implications given the association of *MYC/MYCN* amplification with treatment-refractory disease and relapse in medulloblastoma [34,35]. The same effect was not observed in HD-MB03 cells following YB-1 depletion, likely due to the *MYC*-amplified status of this line, which may act to mask such changes in MYC activation. Interestingly, upstream regulator analysis also revealed inhibition of rapamycin-insensitive companion of mTOR (RICTOR) and regulatory-associated protein of mTOR (RPTOR)—components of the mechanistic target of rapamycin (mTOR) signalling. Previous studies have demonstrated clear communication between YB-1 and the PI3K/AKT/mTOR pathway and thus this finding may indicate a similar association within paediatric brain tumours [36,37,38]. Taken together, these results shed light on the YB-1 transcriptome in MB_Group3_, revealing potential novel functions in cellular stress response, lipid metabolism, as well as MYC and mTOR pathway activity.

### 3.4. YB-1 Inhibition Is Associated with Increased Cellular Sensitivity to Vincristine, Partly through Reduced Expression of ABCB1

YB-1 appears to be implicated in cell death and survival processes and cellular stress mechanisms (Figure 3A–F). This, combined with the well-documented role of YB-1 in multi-drug resistance in various cancer types, led us to explore the association between YB-1 expression and cellular sensitivity to anti-cancer therapies in medulloblastoma cells. YB-1 knockdown cell lines were exposed to standard-of-care chemotherapies vincristine and cisplatin, as well as two novel anti-cancer therapies which have shown promise in medulloblastoma pre-clinical studies, JQ1 (a BET bromodomain inhibitor) and panobinostat (a histone deacetylase inhibitor). YB-1 depletion was found to result in increased sensitivity to vincristine in KD-D283 cells and KDHD-MB03 cells compared to non-silencing controls (Figure 4A,B). Interestingly, this falls within the therapeutic concentration range of vincristine in the cerebrospinal fluid (CSF), indicating that genetic inhibition of YB-1 can potentiate vincristine treatment within a clinically achievable range in both cell lines [26]. Similarly, YB-1 knockdown promoted a ~15–20% increase in cell sensitivity to panobinostat and a ~10% increase in cell sensitivity to JQ1 across a broad concentration range in the KD-HD-MB03 cell line. Comparatively, YB-1 knockdown had no effect on cell sensitivity to cisplatin in either cell line, or to JQ1 or panobinostat in the KD-D283 cell line and did not impact the cell line proliferation rate (Figure 2F).

Vincristine is a well-established ABCB1 transport substrate and the increased sensitivity to vincristine in both knockdown cell lines led us to further interrogate the YB-1-ABCB1 axis in MB_Group3_ [27]. ChIP assays revealed substantial YB-1 enrichment at a previously unknown inverted CCAAT box motif within the *ABCB1* exon 3 promoter region within both cell lines (Figure 4C,D). To validate this finding, *ABCB1* mRNA expression was assessed in YB-1 knockdown cell lines and non-silencing control lines by qRT-PCR. As anticipated, genetic disruption of YB-1 significantly reduced the expression of *ABCB1* mRNA by ~40% in HD-MB03 cells and ~50% in D283 cell lines, further supporting the notion that YB-1 transcriptionally regulates *ABCB1* in medulloblastoma cells (Figure 4E). Taken together, these data suggest that YB-1 expression is required for the maintenance of intrinsic cellular resistance to certain anti-cancer agents, both through the transcriptional regulation of multi-drug transporter pump *ABCB1* and in the case of cisplatin and JQ1, two non-ABCB1 substrates, alternative resistance mechanisms.

### 3.5. Drug-Tolerant Medulloblastoma Cell Lines Exhibit a Common Gene Signature Associated with Chemoresistance

Since YB-1 knockdown did not potentiate cisplatin, a further, non-targeted approach was utilised to identify drug resistance mechanisms independent from YB-1 and the YB-1-ABCB1 axis. As such, in vitro models of stable, acquired drug tolerance (DT) were generated utilising standard-of-care medulloblastoma chemotherapy agents: cisplatin and vincristine. Following continuous drug treatment and dose escalation following cell proliferation, three cisplatin-tolerant MB_Group3_ cell lines DT-D283-CIS, DT-D458-CIS and DT-HD-MB03-CIS were derived, with significantly elevated IC_50_ values compared to the respective parental lines D283, D458 and HD-MB03 (2.5-, 18.5- and 1.6-fold increase in cisplatin resistance respectively; Figure 5A–C). No MB_Group3_ cell line tested was able to develop tolerance to vincristine, with dose escalation associated with cellular senescence and death (Louisa Taylor, Ian Kerr & Beth Coyle, University of Nottingham, UK, observation, 2021). Contrastingly, MB_SHH_ cell line DAOY did develop vincristine tolerance (DT-DAOY-VIN), displaying a significant 2.8-fold increase in IC_50_ compared to the parental line (Figure 5D). It is possible that this disparity between MB_SHH_ and MB_Group3_ lines lies in the differences between *ABCB1* expression between cell lines associated with these groups. DAOY cells express approximately 15-fold more *ABCB1* than the MB_Group3_ cell lines included in this study (Appendix A). Due to the capacity of vincristine to act as a substrate of ABCB1, such increased expression may continue to facilitate drug export and cell survival even during chronic vincristine exposure.

Assessment of ABC transporter expression supported this theory, with the DT-DAOY-VIN line exhibiting significantly elevated expression of *ABCB1* compared to its vehicle-treated control sister line (DT-DAOY-DMSO; Appendix A). Elevated expression of *ABCC1*, of which vincristine is also a substrate, was also detected in DT-DAOY-VIN (Appendix A). Interestingly, significantly increased *ABCC1* was also recorded in the DT-D458-CIS and DT-D283-CIS lines compared to vehicle-control lines (DT-D458-DMF and DT-D283-DMF), despite cisplatin being a non-ABCC1 substrate; perhaps indicative of a function of ABCC1 in therapy resistance and/or cell survival independent of its role in cytotoxic drug efflux [28].

The aforementioned drug-tolerant cell lines underwent 3′mRNA sequencing to elucidate transcriptomic alterations that occur following chronic drug exposure and the concurrent development of drug tolerance. Unsupervised hierarchical clustering analysis revealed significant alteration in the transcriptome following the acquisition of drug tolerance (Appendix A). Interestingly, the number of significantly differentially expressed genes identified appeared to correlate with the level of acquired resistance, with the DT-D458-CIS line, which displayed an 18.5-fold increase in cisplatin resistance, exhibiting the highest number of differentially expressed genes (2148 DEGs) and the DT-HD-MB03-CIS line, which displayed a 1.6-fold increase in cisplatin resistance, exhibiting the fewest (910 DEGs) (Appendix A).

Comparison of significant DEGs between drug-tolerant cell lines displayed a high level of overlap. 59 genes were commonly up-regulated and 28 genes were commonly down-regulated in at least three of the four drug-tolerant cell lines compared to the appropriate vehicle-treated controls (Figure 5E,F). These were predominately shared between the cisplatin-tolerant MB_Group3_ lines (27 commonly upregulated genes and 12 commonly down-regulated genes, Figure 5E,F), indicative of common gene alterations following the acquisition of cisplatin resistance in medulloblastoma cells. Most strikingly, our analysis revealed a seven-gene drug-tolerant signature in all four drug-tolerant cell lines, irrespective of chemotherapeutic treatment, parental cell line and tumour subgroup—*LTBP1*, *MAP1A*, *MBNL2*, *LGALS1*, *PNRC1*, *DAB2* and *PLAAT3* (Figure 5E). It is feasible that these seven-shared genes may be associated with global mechanisms of drug tolerance, common to multiple treatment types. Examination of the expression of these seven genes within a publicly available small-scale medulloblastoma dataset containing diagnosis, post-treatment and relapse patient samples supported this theory [18]. *MAP1A*, *MBNL2* and *PLAAT3* gene expression were found to be significantly elevated in post-treatment patient samples, while *LTBP1* expression was found significantly increased in relapse samples, in comparison to samples taken at diagnosis (Figure 5G,H and Appendix A). Thus, there exists here the potential to uncover novel, targetable therapeutic hits implicated in the acquisition of resistance to standard medulloblastoma therapies. Future clinical validation on a larger scale will be key to further exploring this exciting finding.

## 4. Discussion

Medulloblastoma survival by conventional treatment regimens currently lies at 60%. This figure drops drastically upon tumour relapse, which occurs in 30% of patients and is almost universally fatal. A key hurdle to effective medulloblastoma treatment is cellular resistance to standard-of-care therapies. In this report, we identified mechanisms surrounding medulloblastoma therapy resistance, both related to and independent from oncoprotein YB-1. In the present study, we show, for the first time, that high YB-1 gene expression is associated with poor survival outcomes in MB_SHH_, MB_Group3_ and MB_Group4_ patients, providing evidence of a likely oncogenic function for the protein in medulloblastoma. Accordingly, previous research has identified a role for YB-1 in driving the proliferation of both MB_SHH_ cells and cerebellar granule neural precursor cells [29]. YB-1 also appears to indirectly regulate the transcription of genes implicated in cell death and the medulloblastoma inflammatory response [39].

To further investigate the functional role of YB-1 in medulloblastoma, we generated YB-1 knockdown MB_Group3_ cell lines. We found that YB-1 depletion results in diminished cell invasion in a modified transwell invasion assay model, in a manner independent from MMP2 and MMP9 activity. This is supported by previous work, in which YB-1 has been linked to tumour cell invasion in both epithelial and brain tumour models and highlights, for the first time, an association between YB-1 and invasion in medulloblastoma [40,41,42,43]. As well as identifying extracellular matrix components and cell adhesion proteins as candidates for YB-1 mediated invasion, whole transcriptome sequencing of YB-1 knockdown MB_Group3_ medulloblastoma cell lines identified a number of pathways and upstream regulators within cell death and survival and lipid metabolism cellular functions, which was further validated using publicly medulloblastoma patient cohorts. Especially notable given its role in very high-risk and treatment-refractory medulloblastoma, was the significant inactivation of the MYC oncoprotein following YB-1 knockdown in the D283 MB_Group3_ cell line and medulloblastoma patients with low YB-1 gene expression [3]. In contrast to previous reports, MYC protein and mRNA expression were unaltered in YB-1 knockdown cells, indicative that the observed effect occurs by alteration in MYC activity rather than expression [44,45]. Numerous studies support a model in which MYC interacts differentially with various co-factors in dynamic complexes [46]. It is feasible that YB-1 represents one such co-factor in D283 cells, as supported by a proteomic profiling study conducted by Agrawal et al., in which YB-1 was identified as an MYC-interacting protein [47].

We took both targeted and non-targeted approaches to investigate mechanisms of resistance to standard-of-care therapeutics in medulloblastoma cell lines. Previously, we showed ABCB1 expression to be associated with high-risk medulloblastoma [48]. Building upon this, in the current study we demonstrated that a YB-1-ABCB1 axis mediated MB_Group3_ cellular sensitivity to vincristine. ChIP analysis showed YB-1 enrichment at a hitherto unidentified inverted CCAAT box site within the *ABCB1* exon 3 promoter region, and this was supported by a significant reduction in *ABCB1* mRNA expression in YB-1 knockdown cells. Although the level at which YB-1 regulates *ABCB1* expression has been debated [49,50], our results are in agreement with previous studies which have highlighted a probable role for YB-1 in *ABCB1* transcriptional control, displaying interaction at alternative Y-box and CCAAT sites within the exon 1 and exon 3 ABCB1 promoter regions, respectively [51,52]. Intriguingly, as well as the transcriptional regulation of *ABCB1*, our transcriptomic analysis of YB-1 knockdown cell lines identified dysregulation of lipid and cholesterol biosynthesis pathways. Given that the functional activity of ABCB1 is modulated by the composition of the lipid bilayer, and in particular the presence of cholesterol [53,54,55,56], it is tempting to speculate that the observed effect on vincristine sensitivity in YB-1 knockdown cell lines is mediated both by YB-1 depletion and by YB-1-driven dysregulation of lipid metabolism processes. Such dysregulation of ABCB1 may also relate to our previously discussed observations regarding diminished cell invasion following YB-1 knockdown. Indeed, ABCB1 protein expression is associated with metastatic medulloblastoma and inhibition of ABCB1 impedes the migratory capacity of medulloblastoma cells [57]. It is feasible, therefore, that in addition to vincristine sensitivity, YB-1-mediated ABCB1 expression and function contribute to the invasive capabilities of medulloblastoma cells.

The re-programming of lipid metabolism represents a newly recognised hallmark of cancer, with lipid and cholesterol biosynthesis frequently exploited in cancer cells to meet energy demands for rapid growth [58]. In support of our findings in both cell lines and patient datasets, an association between YB-1 and mono-unsaturated fatty acid biosynthesis has been previously demonstrated [59]. Furthermore, in vivo models of MB_SHH_ display exaggerated lipogenesis, with lipidome alterations detectable between in vivo models of non-metastatic and metastatic MB_SHH_, indicative of concordance between lipid metabolism and medulloblastoma tumour progression and highlighting the YB-1-lipidome interaction as an intriguing area for further study [60,61].

YB-1 independent mechanisms of drug resistance were revealed by 3′mRNA sequencing of three cisplatin-tolerant MB_Group3_ cell lines and one vincristine-tolerant MB_SHH_ cell line. The most striking observation was the existence of a seven-gene drug-tolerant signature in all four drug-tolerant cell lines sequenced, irrespective of chemotherapeutic treatment or medulloblastoma group cell line. Of these seven genes (*LTBP1*, *MAP1A*, *MBNL2*, *LGALS1*, *PNRC1*, *DAB2*, and *PLAAT3*), only one, *LGALS1*, had been researched previously in relation to medulloblastoma, where it is up-regulated in MB_SHH_ patients and activated by the SHH signalling pathway [62]. Thus, there exists here the potential to uncover and characterise novel therapeutic targets implicated in the acquisition of resistance to cisplatin and vincristine in medulloblastoma. Accordingly, characterisation and clinical validation of this seven-gene drug-tolerant signature represent an important area of future research.

## 5. Conclusions

In conclusion, we demonstrate here an association between YB-1 expression and various aspects of medulloblastoma tumourigenesis, including cell invasion, MYC oncoprotein activity and lipid metabolism. Bioinformatic analysis of published datasets validated these pre-clinical findings, supporting a role for YB-1 in cell death and survival processes, cholesterol biosynthesis, response to therapeutics and MYC/mTOR signalling; as well as overall survival and metastasis in medulloblastoma patients. Importantly, we also identify YB-1 as a transcriptional regulator of drug resistance-related gene *ABCB1* and provide evidence to support a function for YB-1 in intrinsic cellular resistance to various anti-cancer therapies, both related to and separate from its aforementioned role as a regulator of the *ABCB1* gene. Finally, through the establishment and next-generation sequencing of drug-tolerant medulloblastoma cell lines, we identify targetable hits implicated in the acquisition of tolerance to standard-of-care agents, which may inform future pre-clinical investigations with a focus on hindering the development of therapy resistance.

## Figures and Tables

**Figure 1 cancers-15-01086-f001:**
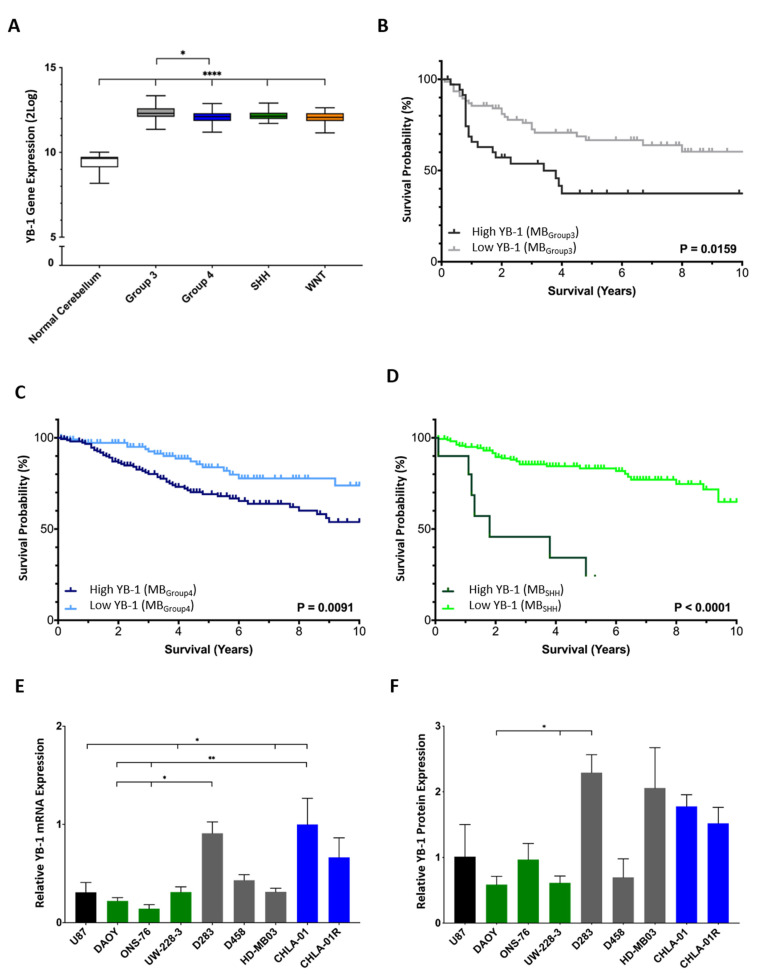
YB-1 is expressed highly in medulloblastoma and is associated with poor prognosis. (**A**) YB-1 gene expression is elevated in all four medulloblastoma principal groups. Normal cerebellum *n* = 9; MB_Group3_ *n* = 56, MB_Group4_ *n* = 91, MB_SHH_ *n* = 59 and MB_WNT_ *n* = 17. Expression displayed as box plots showing the sample minimum (lower line), lower quartile (bottom of box), median (line within box), upper quartile (top of box) and the sample maximum (upper line). (**B**–**D**) Kaplan-Meier analysis revealed high YB-1 expression is associated with poor 10-year overall survival in MB_Group3_ ((**B**); *n* = 144), MB_Group4_ ((**C**); *n* = 326) and MB_SHH_ ((**D**); *n* = 223) patients. Survival curves were compared using the Log-rank (Mantel-Cox) test. (**E**) Analysis of YB-1 mRNA expression by qRT-PCR revealed YB-1 expression across all available medulloblastoma cell lines. Gene expression was calculated relative to housekeeping gene *GAPDH* (ΔCq). U87 cells were utilised as a positive control. (**F**) Densitometry analysis of YB-1 protein expression relative to GAPDH expression revealed that YB-1 was expressed in all medulloblastoma cell lines examined. Cell lines are colour-coded as MB_SHH_ (green), MB_Group3_ (grey) and MB_Group4_ (blue). * *p* < 0.05, ** *p* < 0.01; **** *p* < 0.0001. Only comparisons where there was statistical significance are shown. All other pairwise comparisons are non-significant.

**Figure 2 cancers-15-01086-f002:**
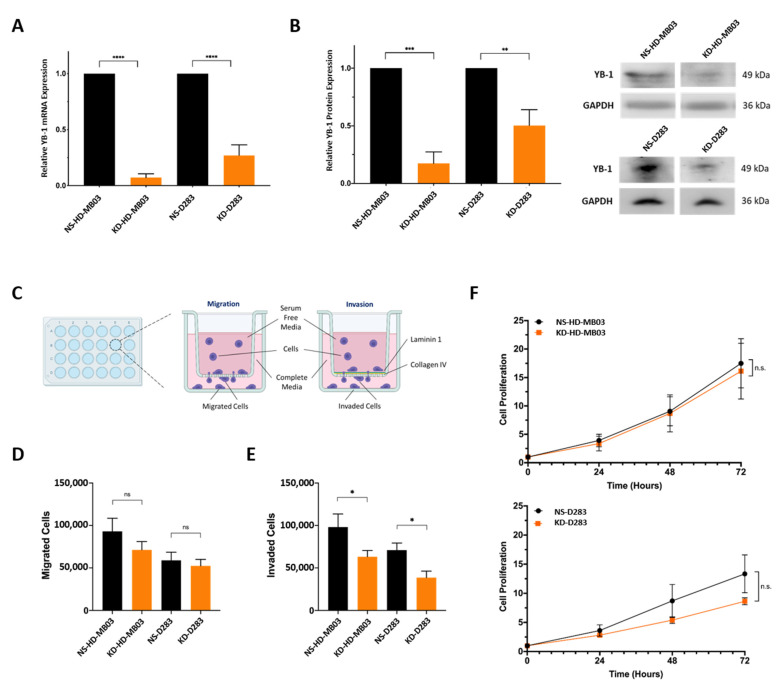
YB-1 knockdown inhibits Group 3 medulloblastoma cell invasion. (**A**) In HD-MB03 and D283 cells, YB-1 was significantly depleted using the Y_A shRNA construct at an MOI of 0.1. Relative YB-1 mRNA expression displayed as fold change (2^−ΔΔCq^) relative to the appropriate non-silencing control. *n* = 3; mean ± SEM. Significance was assessed by ordinary one-way ANOVA analysis with Sidak’s multiple comparison tests. (**B**) Western blot analysis and concurrent densitometry revealed YB-1 protein expression to be significantly depleted in the KD-HD-MB03 and KD-D283 lines. Densitometry data are presented relative to the GAPDH loading control and normalised to the appropriate non-silencing control cell line. *n* = 3; mean ± SEM; ** *p* < 0.01, *** *p* < 0.001, **** *p* < 0.0001. Significance was assessed by ordinary one-way ANOVA analysis with Sidak’s multiple comparisons tests. (**C**) A schematic representation of the transwell migration and invasion assays was utilised in this experiment. Medulloblastoma cells were seeded in serum-free media and migrated (uncoated insert) or invaded (laminin 1/collagen IV-coated insert) for 48 h, facilitated by an FBS gradient. (**D**) YB-1 depletion did not have a significant effect on the migratory capacity of either KD-HD-MB03 or KD-D283 compared to the appropriate non-silencing control cell line. (**E**) YB-1 depletion resulted in a significant reduction in the number of invading cells detected in both KD-HD-MB03 and KD-D283 compared to the appropriate non-silencing control cell line. Mean ± SEM plotted; *n* = 3; * *p* < 0.05, n.s. = not significant; significance assessed by unpaired *t*-test. (**F**) To assess the effect of YB-1 knockdown on proliferation, cell metabolic activity was assessed at 0, 24, 48 and 72 h time points using an end-point PrestoBlue cell viability assay. No difference in relative cell proliferation was detected between non-silencing and YB-1 knockdown cells in either cell line. *n* = 3 (HD-MB03); *n* = 5 (D283). Metabolic activity at each time point relative to metabolic activity at the 0 h time point was used to calculate relative cell proliferation. Mean ± SEM plotted. Significance was assessed by way of multiple *t*-tests with Holm-Sidak multiple comparisons testing. The uncropped blots are shown in Appendix A.

**Figure 3 cancers-15-01086-f003:**
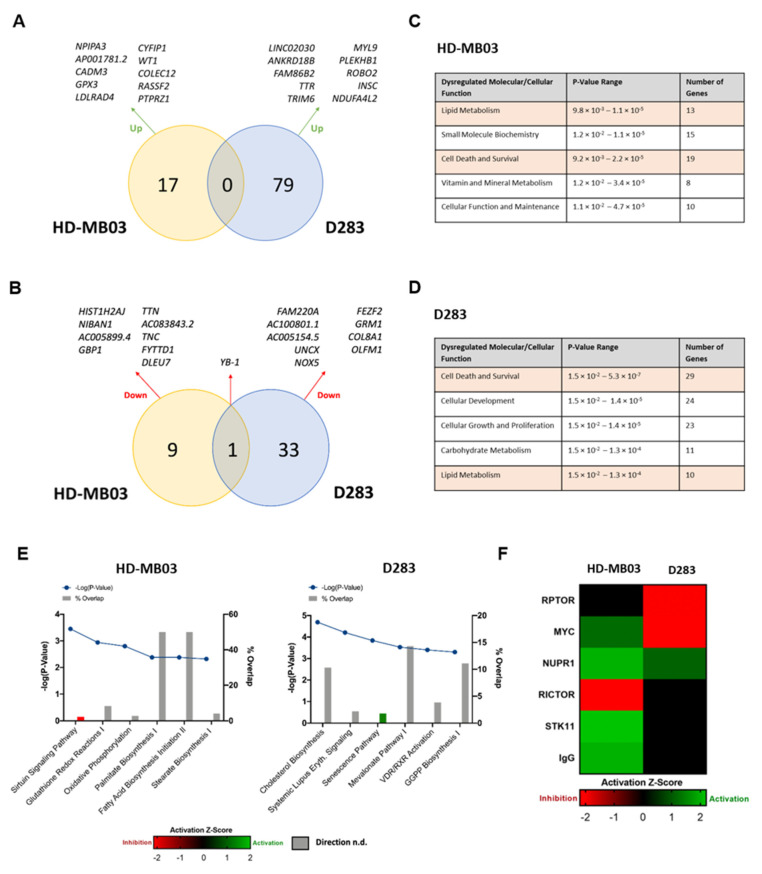
Whole transcriptome sequencing of YB-1 knockdown cell lines demonstrates roles for YB-1 in key cellular pathways in medulloblastoma cells. (**A**) Venn diagram displaying the number of significantly up-regulated genes following YB-1 knockdown in KD-HD-MB03 and KD-D283 lines, with the top 10 most significant genes highlighted. (**B**) Venn diagram displaying the number of significantly down-regulated genes following YB-1 knockdown in KD-HD-MB03 and KD-D283 lines, with the top 10 most significant genes highlighted. Genes were ordered based on Log2 fold change. Significance was determined using an uncorrected *p*-value (≤0.05) and an FDR *p*-value (≤0.1). *p*-Values were adjusted using the Benjamini-Hochberg False Discovery Rate (FDR) approach to correct for multiple testing. (**C**) The 5 most significantly dysregulated functions following YB-1 knockdown in HD-MB03 and D283 (**D**) cells. (**E**) Top significantly altered canonical pathways following YB-1 knockdown in HD-MB03 and D283 cells are displayed. Where possible, an activation Z-score was determined, red indicates an inhibitory effect while green indicates an activational effect. Grey bars represent pathways for which a Z-score could not be determined (n.d.). Significance was calculated using Fisher’s exact test. (**F**) Heatmap presenting results of Upstream Regulator Analysis. mTOR binding protein RICTOR was found to be significantly inhibited in KD- HD-MB03 cells (Z-score = −2.2), while MYC and mTOR binding protein RPTOR were found to be significantly inhibited in KD-D283 cells (Z-scores = −2.1 and −2.0 respectively).

**Figure 4 cancers-15-01086-f004:**
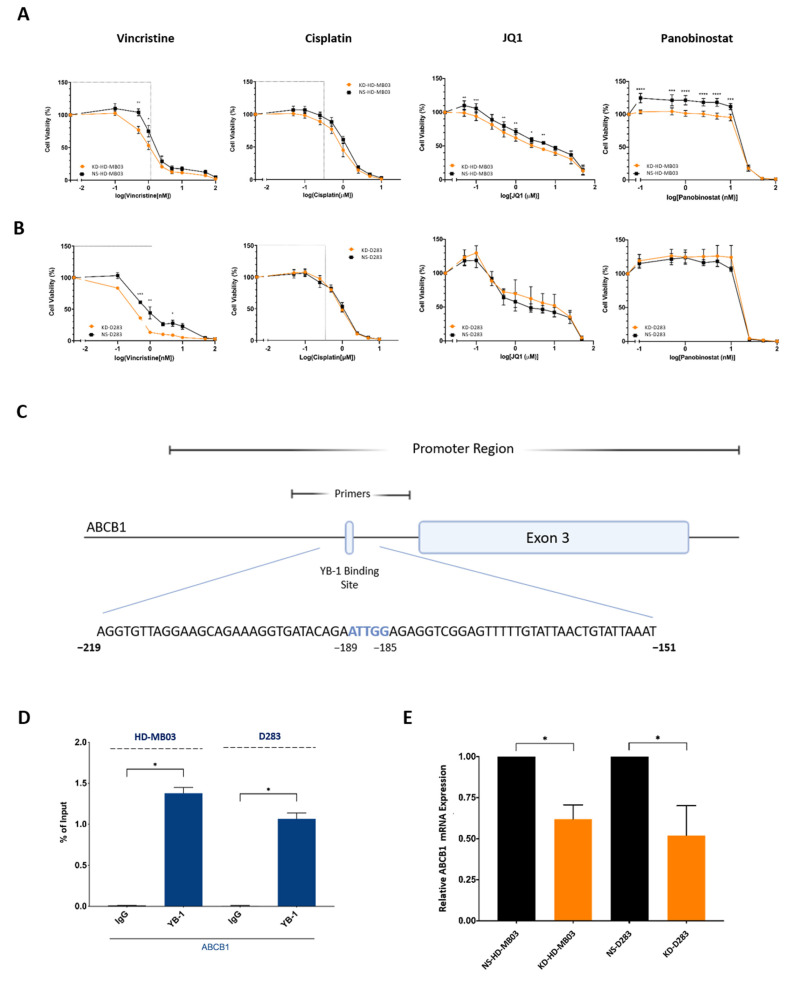
YB-1 depletion alters medulloblastoma cell sensitivity to various anti-cancer therapies. KD/NS-D283 and KD/NS-HD-MB03 cells were treated with vincristine (0.1–100 nM), cisplatin (0.05–10 µM), JQ1 (0.025–50 µM) and panobinostat (0.1–100 nM) for 72 h, after which cell viability was assessed by PrestoBlue metabolic assays. (**A**) KD-HD-MB03 cells showed a significant reduction in cell viability compared to the NS-HD-MB03 control line in response to vincristine (0.5–1 nM), panobinostat (0.1–10 nM) and JQ1 (0.05, 0.1, 0.5, 1.0 and 5 µM) treatment). (**B**) KD-D283 cells showed a significant reduction in cell viability compared to the NS-D283 control line at 0.5 nM, 1 nM and 5 nM vincristine treatment. Grey boxes depict clinically achievable drug CSF concentrations. Mean ± SEM plotted; *n* = 4. Significance was assessed by way of Two-Way ANOVA with Sidak’s multiple comparison tests. (**C**) Schematic displaying the identified YB-1 binding site (B.S.) in the *ABCB1* promoter region. The core inverted CCAAT pentanucleotide is highlighted in blue. (**D**) qPCR analysis of ChIP assay products revealed that YB-1 binds strongly to an inverted CCAAT box in the *ABCB1* promoter of HD-MB03 and D283 cell lines. *n* = 3; data normalised to input; mean ± SEM plotted; significance assessed by paired *t*-test analysis. (**E**) *ABCB1* expression in KD-D283 and KD-HD-MB03 cell lines was analysed by qRT-PCR and quantified relative to non-silencing controls (2^−ΔΔCq^). KD-HD-MB03 (*n* = 3) and KD-D283 cells (*n* = 5) exhibited a significant reduction in *ABCB1* mRNA expression relative to NS-HD-MB03 and KD-D283. Mean ± SEM plotted; significance assessed by unpaired *t*-test. Significant differences indicated as * *p* < 0.05, ** *p* < 0.01, *** *p* < 0.001 and *** *p* < 0.0001.

**Figure 5 cancers-15-01086-f005:**
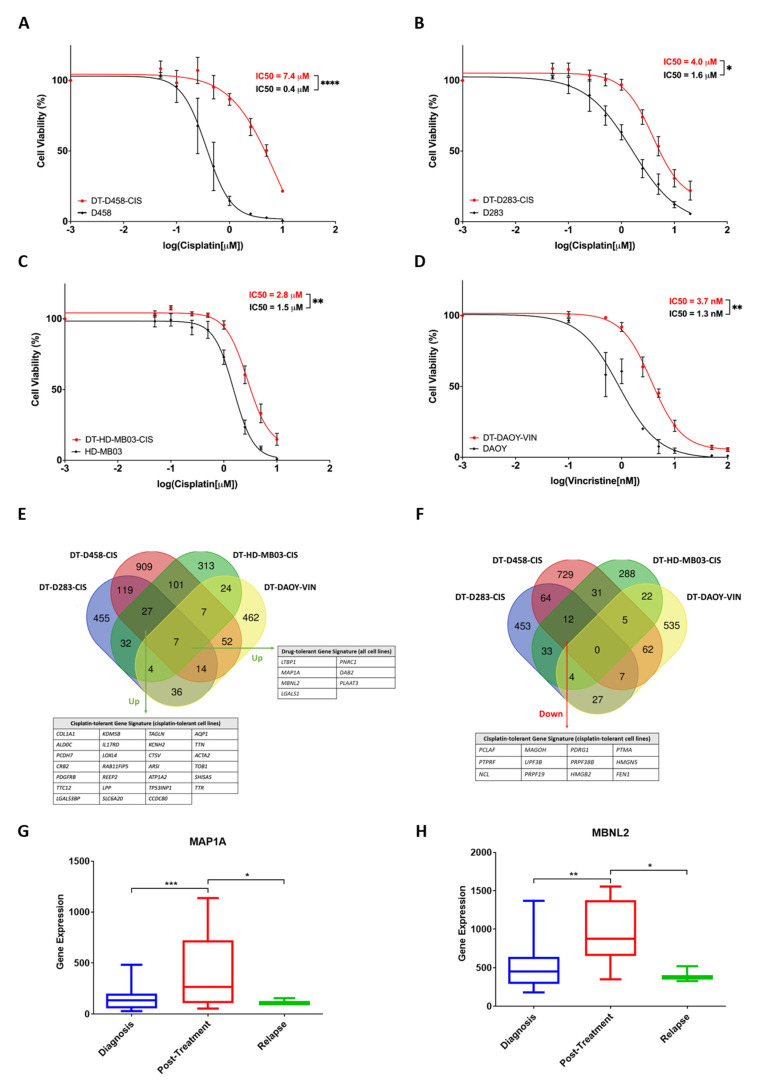
3′mRNA-Seq of drug-tolerant medulloblastoma cell lines reveals a gene signature associated with cisplatin and vincristine resistance. (**A**) D458, D283 and HD-MB03 cell lines were cultured continuously in the presence of cisplatin. Cell viability assays (PrestoBlue) allowed monitoring of cellular resistance compared to a vehicle-treated control line and/or the parental line. DT-D458-CIS cells exhibited an 18.5-fold increase in IC_50_ value compared to the parental cell line following continuous treatment in 0.6 µM cisplatin. (**B**) DT-D283-CIS cells exhibited a 2.5-fold increase in IC_50_ value compared to the parental cell line following continuous treatment in 1.6 µM cisplatin. (**C**) DT-HD-MB03-CIS cells exhibited a 1.6-fold increase in IC_50_ value compared to the parental cell line following continuous treatment in 0.5 µM cisplatin. (**D**) DT-DAOY-VIN cells were cultured continuously in the presence of vincristine. Cells exhibited a 2.8-fold increase in IC_50_ value compared to the parental cell line following continuous treatment with 2 nM vincristine. Mean ± SEM plotted; *n* = 3. Dose-response curves were generated using non-linear regression analyses and IC_50_ values were calculated accordingly. The significance of IC_50_ values was assessed by ordinary one-way ANOVA analyses with Tukey’s multiple comparisons tests. (**E**) Significantly up-regulated genes and down-regulated genes (**F**) were detected in drug-tolerant cell lines compared to vehicle-treated control cell lines. Significantly up-/down-regulated genes were those found to be up-regulated in DT cell lines compared to vehicle-treated control lines, with a Log2 fold change of ≥+0.5/≤−0.5, a BH-adjusted *p*-value of ≤0.05 and expression of ≥2 normalised counts. 7 genes were identified as commonly up-regulated between all four DT cell lines. (**G**) *MAP1A* and *MBNL2* (**H**) expression is elevated in medulloblastoma patient samples collected post-therapy compared to samples collected at diagnosis or relapse. Diagnosis *n* = 46; post-treatment *n* = 8, relapse *n* = 3. Expression is displayed as box plots showing the sample minimum (lower line) and the sample maximum (upper line). Dataset (Tumor Medulloblastoma public—Delattre—57—MAS5.0—u133p2) accessed using R2: Genomics Analysis and Visualization Platform. * *p* < 0.05, ** *p* < 0.01, *** *p* < 0.001, **** *p* < 0.0001.

## Data Availability

The 3′mRNA Sequencing data of the drug tolerant lines have been deposited in the ArrayExpress database at EMBL-EBI (www.ebi.ac.uk/arrayexpress) under accession number E-MTAB-12686.

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
