# Peer review of "Drug Resistance in Medulloblastoma Is Driven by YB-1, ABCB1 and a Seven-Gene Drug Signature"

_cancers, 2023, doi:10.3390/cancers15041086_

Round 1

Reviewer 1 Report

The study of L.Taylor et al. is a well done investigation of meduloblasoma molecular pathogenesis in the respect of chemoresistance. It demonstrates that (1) high YBX1 expression correlates with poor overall survival in medulloblastoma; (2) YB-1 depletion impedes the invasive capability of medulloblastoma cells; (3) YB-1 is involved in numerous key cellular pathways in medulloblastoma cells; (4) YB-1 inhibition is associated with increased cellular sensitivity to vincristine, partly through reduced expression of ABCB1; (5) drug-tolerant medulloblastoma cell lines exhibit a common 7 gene signature associated with chemoresistance. However, the presentation of the study should be improved to become more understandable and it requires some work to satisfy the style of JIMS.

1. The abstract should be a total of about 200 words maximum. It should follow the style of structured abstracts, but without headings: 1) Background: Place the question addressed in a broad context and highlight the purpose of the study; 2) Methods: Describe briefly the main methods or treatments applied. Include any relevant preregistration numbers, and species and strains of any animals used. 3) Results: Summarize the article's main findings; and 4) Conclusion: Indicate the main conclusions or interpretations.

2.     Introduction should be rewritten to clarify goals and aims of the study, its logic structure, the approaches and methods used for the study fulfilment. Bioinformatics and experimental parts of the study should be described in details so that to show how authors use published datasets and their experimental data.

3.Methods should be described more accurate (in particular, culture conditions do not include antibiotics, CO2, micoplasma control, etc., however CO2 is pointed out in cytotoxicity assays), bioinformatics approaches should be described separately from Statistical Analysis. All the subsections of Materials and Methods should be checked thoroughly.

4.     In the section of Results, it should be clarified when bioinformatics results are presented and where author present their experimental data, what dataset were analyzed in bioinformatics analyses, what approaches was used and what results were obtained. Now it is difficult to understand what means Group 3, there is no any description of it previously, it is not sufficient to point out the reference 14 or 13, as authors present their separate investigation and it should be clear from their own description. Thus, Results also should be carefully written. Moreover, all the figures should be thoroughly checked to point out more understandable axis meaning and correspondence of figures to the legends provided (on the figure 4 I did not find “B”, but it exists in the legend)

5.     In the Discussion authors should add why the only ABCB1 was analyzed, although chemoresistance may involve other transporters.

6.     Conclusion should be added more clear description what was found by bioinformatics analysis of the published datasets, how it was extended by experimental work and then how this experimental results correspond the published datasets.

Author Response

  1. The abstract should be a total of about 200 words maximum. It should follow the style of structured abstracts, but without headings: 1) Background: Place the question addressed in a broad context and highlight the purpose of the study; 2) Methods: Describe briefly the main methods or treatments applied. Include any relevant preregistration numbers, and species and strains of any animals used. 3) Results: Summarize the article's main findings; and 4) Conclusion: Indicate the main conclusions or interpretations.
    1. We have reduced the word count and re-arranged the text to better follow the structured abstract style.

  1. Introduction should be rewritten to clarify goals and aims of the study, its logic structure, the approaches and methods used for the study fulfilment. Bioinformatics and experimental parts of the study should be described in details so that to show how authors use published datasets and their experimental data.
    1. Please see lines 77 – 90 for expanded section on study aims and methods used to achieve said aims. The use of published datasets to validate key findings has also now been made clear.

  1. Methods should be described more accurate (in particular, culture conditions do not include antibiotics, CO2, micoplasma control, etc., however CO2 is pointed out in cytotoxicity assays), bioinformatics approaches should be described separately from Statistical Analysis. All the subsections of Materials and Methods should be checked thoroughly.
    1. We thank Reviewer 1 for highlighting this oversight. We have added the required detail to cell culture and in vitro assays. We have also included an additional “Bioinformatic Analysis of Published Datasets” section (line 234) to make clear our use of published datasets and highlight how they were accessed through the R2: Genomics Analysis and Visualization Platform.

  1. In the section of Results, it should be clarified when bioinformatics results are presented and where author present their experimental data, what dataset were analyzed in bioinformatics analyses, what approaches was used and what results were obtained. Now it is difficult to understand what means Group 3, there is no any description of it previously, it is not sufficient to point out the reference 14 or 13, as authors present their separate investigation and it should be clear from their own description. Thus, Results also should be carefully written. Moreover, all the figures should be thoroughly checked to point out more understandable axis meaning and correspondence of figures to the legends provided (on the figure 4 I did not find “B”, but it exists in the legend)
    1. We have now referred to all patient datasets referenced in the study as “published” to ensure it is transparent when we are referring to publicly available datasets and when we are referring to our own experimental data. This is further clarified in the methods (See point 3).
    2. We have added a description of the medulloblastoma principal groups to the introduction to define WNT, SHH, Group 3 and Group 4 and ensured it is clear which cell line belongs to which group.
    3. We have checked to make sure all legends correspond to figures and made sure all axis titles are consistent.

  1. In the Discussion authors should add why the only ABCB1 was analyzed, although chemoresistance may involve other transporters.
    1. Thank you for this comment. We have now added (line 591) that the YB-1-ABCB1 axis research in the current study builds upon previous research from our group where we examined the association between ABCB1 and high-risk medulloblastoma.

  1. Conclusion should be added more clear description what was found by bioinformatics analysis of the published datasets, how it was extended by experimental work and then how this experimental results correspond the published datasets.
    1. We have now expanded the role of bioinformatics analysis of published datasets in validating our pre-clinical data to address this important point.

Reviewer 2 Report

In this manuscript by Taylor L et al., the authors have studied the effect of the multi-functional transcription factor YB-1 and its regulation of multi-drug pump ABCB1 in drug resistance in medulloblastoma cell lines. Through cell migration and cell invasion studies they have shown the role of YB1 in metastasis. Moreover, through RNA seq analysis and Chromatin Immunoprecipitation they have found the role of the multidrug resistance transporter ABCB1 in YB1 regulated drug resistance.

This is an interesting study on the topic of drug resistance which is an important problem in the field of cancer therapeutics. The data representation and statistical analysis needs to be further improved. Moreover, the switch between YB1 and YBX1 impedes the flow of reading. Instead, they can simply mention YB1 mRNA or protein expression. Based on my comments below the overall enthusiasm for publishing this manuscript in the journal ‘Cancers’ is medium to high.

1.     In line 235, the authors should expand and explain the categorization of SHH medulloblastoma patients. Also please specify the correlation of the different categories to disease prognosis in medulloblastoma.

2.     In Figures 1E and 1F, the authors should simply mark the figures as relative mRNA expression (1E) and relative protein expression (1F) for ease of understanding. Moreover, they should provide the western blots for 1F. Why have the authors used U87 GBM cell line as a control if U87 also sows high YB1 RNA and protein expression? The authors should rather use a normal cell line derived from the medulloblastoma cell of origin. The statistical analysis is also confusing and incomplete in Figures 1E and 1F.

3.     Also, in figures 2A, 2B the authors should simply label the cell lines as D283-NS/ D283-KD and HD-MB03-NS/ HD-MB03-KD. Their current nomenclatures are confusing.

4.     It’s surprising that there are no common up or down regulated genes between the HD-MB03 and D283 cell lines. Could the authors comment on this?

5.     Do Figures 3C and 3D show upregulated or downregulated pathways?

6.     In Figure 4A, the authors should also show cell proliferation graph of parental cells which are originally sensitive to Vincristine and their comparison to the paired Vincristine resistant lines.

7.     In Figure 4. The authors should also check ABCB1 protein levels upon YB-1 KD through Immunoblotting.

8.     In Figures 5G,H, does MAP1A and MBNL2 levels go down upon YB1, ABCB1 or YB1+ABCB1 KD?

Author Response

In this manuscript by Taylor L et al., the authors have studied the effect of the multi-functional transcription factor YB-1 and its regulation of multi-drug pump ABCB1 in drug resistance in medulloblastoma cell lines. Through cell migration and cell invasion studies they have shown the role of YB1 in metastasis. Moreover, through RNA seq analysis and Chromatin Immunoprecipitation they have found the role of the multidrug resistance transporter ABCB1 in YB1 regulated drug resistance.

This is an interesting study on the topic of drug resistance which is an important problem in the field of cancer therapeutics. The data representation and statistical analysis needs to be further improved. Moreover, the switch between YB1 and YBX1 impedes the flow of reading. Instead, they can simply mention YB1 mRNA or protein expression. Based on my comments below the overall enthusiasm for publishing this manuscript in the journal ‘Cancers’ is medium to high.

  1. We agree with Reviewer 2’s comment and have now changed YBX1 to YB-1 and mentioned either mRNA or protein to provide a distinction.

  1. In line 235, the authors should expand and explain the categorization of SHH medulloblastoma patients. Also please specify the correlation of the different categories to disease prognosis in medulloblastoma.
    1. SHH, Group 3 and Group 4 patients were all categorised by high-low YB-1 expression which has been expanded in the methods (line 233)
    2. We have clarified the differences in prognosis between the subgroups in the introduction (lines 49-57).
  2. In Figures 1E and 1F, the authors should simply mark the figures as relative mRNA expression (1E) and relative protein expression (1F) for ease of understanding. Moreover, they should provide the western blots for 1F. Why have the authors used U87 GBM cell line as a control if U87 also sows high YB1 RNA and protein expression? The authors should rather use a normal cell line derived from the medulloblastoma cell of origin. The statistical analysis is also confusing and incomplete in Figures 1E and 1F.
    1. We thank Reviewer 2 for these suggestions. We have amended the figures as YB-1 mRNA and YB-1 protein for Figure 1 and any subsequent plots. Data analysis methodology has also now been added to the figure legend.
    2. The western blots for 1F have been provided in Supplementary A3.
    3. The GBM line U87 was selected to be used as a positive control as it is known to express high protein levels of YB-1. This has now been clarified in the figure legend.
    4. To the best of our knowledge, the likely cells of origin for Group 3 and SHH medulloblastoma (Nestin+ cerebellar NPCs and NPCs of the upper rhombic lip, respectively) are not accessible to culture in vitro.

  1. Also, in figures 2A, 2B the authors should simply label the cell lines as D283-NS/ D283-KD and HD-MB03-NS/ HD-MB03-KD. Their current nomenclatures are confusing.
    1. All text and figures have been amended to follow this suggestion.

  1. It’s surprising that there are no common up or down regulated genes between the HD-MB03 and D283 cell lines. Could the authors comment on this?
    1. Please see lines 322 – 324 where we have commented on this finding.

  1. Do Figures 3C and 3D show upregulated or downregulated pathways?
    1. The IPA conducted in figure 3C and 3D allowed us to identify cellular functions that have been significantly dysregulated and encompass a combination of up- and down-regulated genes that will affect different cellular pathways within each function differently. This explanation has now been clarified on the figure.

  1. In Figure 4A, the authors should also show cell proliferation graph of parental cells which are originally sensitive to Vincristine and their comparison to the paired Vincristine resistant lines.
    1. Please see the proliferation assay curves displayed in Figure 2F. We have now referenced these in the text describing Figure 4A (line 437).

  1. In Figure 4. The authors should also check ABCB1 protein levels upon YB-1 KD through Immunoblotting.
    1. Thank you for this suggestion. As we are examining the transcriptional control of ABCB1 by YB-1, we feel that our investigation of ABCB1 mRNA expression is sufficient at the present time.
  2. In Figures 5G,H, does MAP1A and MBNL2 levels go down upon YB1, ABCB1 or YB1+ABCB1 KD?

  1. In Figure 5, we aimed to examine alternative resistance mechanisms in medulloblastoma separate from YB-1 and ABCB1. For this reason we decided to use parental medulloblastoma cell lines rather than the YB-1 KD cell lines utilised in Figures 2 – 4. We also have yet to generate ABCB1-depleted medulloblastoma cell lines, primarily as the majority of selection drugs are ABCB1 substrates. However, work is currently underway in the group to address this using CRISPR-Cas9 with GFP selection.

Round 2

Reviewer 2 Report

In their rebuttal to the manuscript by Taylor L et al., the authors have not satisfactorily responded to most of my comments. Line numbers have been incorrectly provided. Moreover, response to my comments is unsatisfactory.

  1. The line number provided for the response to comment 1a is not correct. Also please expand ‘SHH’ in the classification in lines 57-60.
  2. In Supplementary 1C- GAPDH bands are not visible. Please replace the figure.
  3. The authors have not responded to the statistical analysis question in Comment 2.
  4. In response to comment 3, I do not see the change reflected in the figures.
  5. The line numbers for Comment 4 is also not correctly provided. I could not locate the change in the text. Figure 3 is repeated in the revised manuscript. 
  6. In response to comment 5, the authors should specify in an additional column that which pathways are upregulated, and which ones are downregulated.
  7. In response to comment 6 – Figure 2F does not show cell proliferation of the parental lines in response to Vincristine. The authors should carefully read my comment. Moreover, line number provided is again not correct.

Author Response

We thank the reviewer for their thorough checking of the manuscript and apologise that the line numbers were displaced by a last minute addition to the introduction. We have now made these additional changes and clarifications. Please see the attachment.
